# Early Cervical Cancer Diagnosis with SWIN-Transformer and Convolutional Neural Networks

**DOI:** 10.3390/diagnostics14202286

**Published:** 2024-10-14

**Authors:** Foziya Ahmed Mohammed, Kula Kekeba Tune, Juhar Ahmed Mohammed, Tizazu Alemu Wassu, Seid Muhie

**Affiliations:** 1Department of Software Engineering, College of Engineering, Addis Ababa Science and Technology University, Addis Ababa 16417, Ethiopia; foziya.ahmed@aastu.edu.et (F.A.M.); kula.kakeba@aastu.edu.et (K.K.T.); 2Center of Excellence for HPC and Big Data Analytics, Addis Ababa Science and Technology University, Addis Ababa 16417, Ethiopia; 3Department of Information Technology, College of Computing and Health Informatics, Wolkite University, Wolkite P.O. Box 07, Ethiopia; 4School of Pharmacy, College of Medicine and Health Science, Hawassa University, Hawassa P.O. Box 05, Ethiopia; juhar2161@gmail.com; 5Addis Hiwot General Hospital, Addis Ababa P.O. Box 170023, Ethiopia; tizazu17@yahoo.com; 6Enkoy LLC, Lanham, MD 20706, USA; 7The Geneva Foundation, Silver Spring, MD 20910, USA

**Keywords:** cervical cancer, early diagnosis, precancerous lesions, SWIN Transformer, convolutional neural networks (CNN), colposcopy images, transfer learning, hybrid models, medical image classification, cancer screening, histopathology

## Abstract

**Introduction:** Early diagnosis of cervical cancer at the precancerous stage is critical for effective treatment and improved patient outcomes. **Objective:** This study aims to explore the use of SWIN Transformer and Convolutional Neural Network (CNN) hybrid models combined with transfer learning to classify precancerous colposcopy images. **Methods:** Out of 913 images from 200 cases obtained from the Colposcopy Image Bank of the International Agency for Research on Cancer, 898 met quality standards and were classified as normal, precancerous, or cancerous based on colposcopy and histopathological findings. The cases corresponding to the 360 precancerous images, along with an equal number of normal cases, were divided into a 70/30 train–test split. The SWIN Transformer and CNN hybrid model combines the advantages of local feature extraction by CNNs with the global context modeling by SWIN Transformers, resulting in superior classification performance and a more automated process. The hybrid model approach involves enhancing image quality through preprocessing, extracting local features with CNNs, capturing the global context with the SWIN Transformer, integrating these features for classification, and refining the training process by tuning hyperparameters. **Results:** The trained model achieved the following classification performances on fivefold cross-validation data: a 94% Area Under the Curve (AUC), an 88% F1 score, and 87% accuracy. On two completely independent test sets, which were never seen by the model during training, the model achieved an 80% AUC, a 75% F1 score, and 75% accuracy on the first test set (precancerous vs. normal) and an 82% AUC, a 78% F1 score, and 75% accuracy on the second test set (cancer vs. normal). **Conclusions**: These high-performance metrics demonstrate the models’ effectiveness in distinguishing precancerous from normal colposcopy images, even with modest datasets, limited data augmentation, and the smaller effect size of precancerous images compared to malignant lesions. The findings suggest that these techniques can significantly aid in the early detection of cervical cancer at the precancerous stage.

## 1. Introduction

Cervical cancer remains a major global health issue, ranking as the fourth most common cancer among women worldwide [1]. It is primarily caused by persistent infection with high-risk human papillomavirus (HPV) types. According to the World Health Organization (WHO), there were approximately 660,000 new cases and 350,000 deaths due to cervical cancer in 2022 [1]. The burden of cervical cancer is especially high in low- and middle-income countries, where 94% of the deaths occur, reflecting disparities in access to HPV vaccination, cervical screening, and treatment services [2]. The considerable geographic variation in cervical cancer burden underlines the urgent need for accessible early diagnosis to enable timely and effective interventions [3,4].

Early diagnosis of cervical cancer is essential for improving treatment outcomes and survival rates. Detecting the disease at the precancerous stage allows for less invasive and more effective treatment options, reducing both morbidity and mortality associated with advanced stages [5,6]. Moreover, early diagnosis facilitates timely interventions, such as administering HPV vaccines, which have been shown to significantly reduce cervical cancer incidence [7,8]. The benefits of early detection extend beyond patient outcomes, enabling better resource allocation and a more efficient use of healthcare services [9]. Effective follow-up and management strategies are also important, as they prevent cervical cancer progression [10], improve prognosis, and significantly reduce treatment costs [11,12,13].

Despite its importance, early diagnosis of cervical cancer remains challenging, even in regions with established screening programs [14]. Traditional screening methods like Pap smears and HPV testing have limitations that affect their effectiveness. Pap smears, though widely used, are subject to variability in interpretation and often have lower sensitivity for detecting precancerous lesions [15]. While HPV testing is more sensitive, it can result in high false-positive rates, leading to unnecessary follow-up procedures and psychological distress [16]. These limitations highlight the need for more accurate, reliable, and accessible diagnostic tools to improve early detection and patient outcomes [17].

Recent advancements in deep learning, particularly with models like convolutional neural networks (CNNs) [18] and transformer-based architectures such as the SWIN Transformer [19], have revolutionized medical imaging and significantly improved early cancer detection [20,21]. These technologies can analyze complex patterns in medical images with remarkable accuracy and consistency, often surpassing traditional methods by detecting subtle differences in tissue structures that might be missed by human observers [22,23,24,25].

In this context, a colposcopy image classification using SWIN Transformer and CNN hybrid models is particularly important for improving early cervical cancer detection and treatment, as this addresses the variability and potential misdiagnosis associated with the slow and costly process of manual image interpretation. The SWIN Transformer and CNN hybrid model for a precancerous colposcopy classification combines the strengths of both architectures to deliver more accurate and interpretable results. The CNN component excels at extracting fine-grained, local features such as edges, textures, and shapes from colposcopy images, while the SWIN Transformer’s mechanism of shifted window attention captures both local and global contextual information, modeling long-range dependencies across heterogeneous images. By leveraging both local feature extraction and the global context, the hybrid model generalizes better across diverse datasets, improving classification accuracy despite variability in image quality and patient factors [26]. These complementary features are then integrated and passed through a classification layer to accurately categorize the images. This dual capability is critical for distinguishing subtle differences in tissues, making it particularly effective for classifying precancerous lesions. Integration of the outputs of the hybrid components reduces false positives and negatives, leading to earlier detection and better patient outcomes, a significant advancement in cervical cancer screening [27,28].

The hybrid model’s performance is further enhanced through the use of transfer learning and data augmentation, allowing it to generalize well despite limited data availability [21]. This approach is especially relevant for cervical cancer, where variations in image quality and patient demographics can affect diagnostic outcomes [29]. Advanced techniques like contrastive learning and multi-task learning help the model focus on precancerous lesions while minimizing the impact of confounding factors [30,31]. These hybrid models offer a robust and efficient solution for early diagnosis and treatment planning in cervical cancer prevention especially in resource-limited settings, where cervical cancer is prevalent, offering reliable diagnostic support and enhancing global screening efforts [1].

However, to fully realize these advancements, certain challenges must be addressed. The need for large, annotated datasets to train deep learning models remains a significant barrier, particularly in low-resource settings [29]. Integrating these models into clinical practice also requires rigorous validation and regulatory approval to ensure their safety and effectiveness. Overcoming these obstacles is crucial for implementing AI-driven diagnostic tools in healthcare, ensuring they provide reliable and actionable insights [14]. Additionally, ethical considerations regarding data privacy and algorithmic transparency must be thoroughly addressed to gain public trust and acceptance, which is essential for the widespread adoption of these technologies [12].

The present study leverages transfer learning with SWIN Transformer and CNN hybrid models to overcome the limitations of existing diagnostic methods, further enhancing the accuracy of detecting precancerous lesions. This approach not only promises significant advancements in early cervical cancer detection, potentially leading to better patient outcomes, but also reduces the strain on healthcare systems by minimizing unnecessary follow-up procedures and treatments resulting from false positives [21]. Moreover, this approach’s broader applicability extends to the early detection of other cancers and diseases [32], in addition to the automatic classification of colposcopy images.

Automatic classification in colposcopy plays a key role in enhancing diagnostic accuracy and efficiency. It assists colposcopists by analyzing images in real-time to accurately identify CIN and other precancerous conditions, reducing the risk of misclassifications and ensuring timely treatments. This reduces the likelihood of misdiagnosis and unnecessary interventions due to overdiagnosis [1,30,31]. Furthermore, by providing a standardized, objective analysis, an automatic classification addresses the variability inherent in manual image annotation, ensuring consistent and reliable interpretations through advanced machine learning models like SWIN Transformers and CNNs [19,27]. The hybrid model’s ability to capture both local and global features in colposcopy images is key to achieving accurate tissue classifications and reliable diagnoses across diverse datasets and clinical settings [19,26,27,30].

Overall, cervical cancer remains a significant global health challenge, particularly in regions with limited access to healthcare services. Early diagnosis is essential for improving patient outcomes and reducing mortality. Advanced deep learning and machine learning techniques offer promising solutions to enhance the accuracy and reliability of early diagnostic methods. This study aims to contribute to these efforts by developing an accessible and reliable diagnostic system, with the potential to improve public health outcomes and the quality of life for women worldwide [33]. The findings of this study may also inform policy decisions regarding cervical cancer screening and prevention programs [34].

### 1.1. Related Studies

The application of SWIN Transformer and CNN hybrid models has revolutionized medical image analyses by offering superior performance in segmentation and classification tasks. The hybrid model combines CNN’s local feature extraction capability with the SWIN Transformer’s ability to capture long-range dependencies, making it highly effective for complex medical images like histopathology and radiology scans [19,35]. This architecture has shown remarkable success in lung cancer detection, leveraging CNN for the feature extraction of local tissues and the SWIN Transformer for capturing global contextual information, significantly improving diagnostic accuracy [28,36]. The hybrid model’s ability to fuse these components enhances both the model’s interpretability and its diagnostic power in multimodal datasets [37].

In brain tumor segmentation, the SWIN Transformer and CNN hybrid model was applied to accurately segment tumor boundaries in MRI scans, significantly outperforming traditional CNN-based approaches due to its advanced attention mechanism, which captures the spatial relationships between different regions of the brain [38]. By using the SWIN Transformer, this model efficiently handles the complex structure of brain tissues, leading to improved segmentation accuracy and generalizability across different datasets [39]. Furthermore, its ability to manage high-dimensional medical images without compromising on computational efficiency sets it apart from other architectures [40].

In colorectal cancer screening, the hybrid model demonstrated enhanced polyp detection by integrating CNN’s capacity for fine-grained feature extraction with the SWIN Transformer’s capabilities of capturing the global context. This resulted in a more accurate identification of polyps in colonoscopy images, improving early detection rates [41]. Additionally, the use of attention mechanisms enabled the model to focus on critical areas of the images, reducing false positives and improving overall diagnostic accuracy [28].

For the diagnosis of Alzheimer’s disease, the SWIN Transformer and CNN hybrid model has been employed in the analysis of PET and MRI scans, effectively fusing structural and functional imaging data. This combination allows for a more comprehensive analysis, improving early detection and treatment planning for Alzheimer’s disease. The SWIN Transformer’s attention mechanism enhances the model’s ability to capture subtle changes in brain structure, which is crucial for early-stage diagnosis [37,42]. The model’s performance was validated across large-scale datasets, showing superior accuracy in classifying different stages of Alzheimer’s compared to traditional CNN-based methods [37].

Moreover, the SWIN Transformer and CNN hybrid model has been applied to cardiac and liver image segmentation tasks, where it achieved state-of-the-art results. The model’s ability to learn both local and global dependencies in CT and MRI scans allowed for a more precise segmentation of organs and better detection of pathological features, which is essential in clinical decision making [38,43]. Its application in these tasks underscores its versatility and robustness in handling various medical imaging challenges [35,40].

### 1.2. Organization of This Paper

This paper first introduces the importance of early cervical cancer detection and the limitations of traditional screening methods. It then details the methodology, focusing on the use of SWIN Transformer and CNN hybrid models for colposcopy image classification. This is followed by results that demonstrate the model’s performance and a discussion of the clinical implications, challenges, and future directions for integrating these advanced machine learning techniques into healthcare practice.

## 2. Materials and Methods

### 2.1. Colposcopy Images

The colposcopy image datasets and metadata, comprising 913 images from 200 cases, were obtained from the Colposcopy Image Bank of the International Agency for Research on Cancer (IARC) [44]. The data and metadata were accessed on 16 April 2024. Of the 200 cases, three cases consisting of a total 15 images were excluded due to inconclusive colposcopy and pathology assessments. The remaining 898 colposcopy images from the 197 cases were grouped into normal (*n* = 94 cases), precancerous (*n* = 77 cases), and cancer (*n* = 26 cases) sets based on the colposcopy examination and histopathology findings (Figure 1a,b and Figure 2).

### 2.2. Clinical Criteria for Case Grouping

Cases were meticulously categorized into normal, precancerous, and cancerous based on colposcopy assessments and histopathology findings (Figure 1b and Figure 2), with HPV positivity, the Transformation Zone, and the SWEDE Score as corroborative metrics.
Normal Group:
∘Cases: a total of 94 cases were characterized by normal colposcopy and histopathology findings.∘Key Features:
▪Mostly HPV-negative cases, with some HPV-positive cases.▪Mainly Type 1 Transformation Zones, with some Type 2 and 3.▪Adequate colposcopy samples showing original squamous epithelium, columnar epithelium, and metaplastic squamous epithelium.▪Brown, faintly or patchy yellow, or unknown iodine staining.▪Low SWEDE scores and normal histopathology findings, such as mature squamous epithelium and nabothian cysts.Precancerous Group:
∘Cases: a total of 77 cases with colposcopy and histopathologic findings indicative of Low-grade Squamous Intraepithelial Lesions (LSILs), including CIN1, and High-grade Squamous Intraepithelial Lesions (HSILs), including CIN2 and CIN3.∘LSILs (CIN1):
▪HPV-positive and -negative cases.▪Type 1 or Type 2 Transformation Zone.▪Thin acetowhite epithelium, irregular borders, fine mosaic, and fine punctation on colposcopy.▪Moderate SWEDE scores and faintly or patchy yellow iodine uptake.∘HSILs (CIN2 and CIN3):
▪Higher severity precancerous lesions.▪Dense acetowhite epithelium, sharp borders, ridge sign, inner border sign, and atypical vessels on colposcopy.▪Medium SWEDE scores and distinct yellow iodine uptake.▪HSIL cases required medical intervention to prevent progression to invasive cancer.Cancer Group:
∘Cases: a total of 26 cases with invasive squamous cell carcinoma, adenocarcinoma, and adenocarcinoma in situ.∘Key Features:
▪All HPV-positive cases.▪Type 3 Transformation Zone.▪Colposcopy findings include dense acetowhite epithelium, coarse punctation or mosaic, sharp borders, and features suspicious for invasion like irregular surfaces, erosion, or gross neoplasm.▪High SWEDE scores and distinct yellow or non-staining iodine uptake.▪Histopathology confirmed invasive carcinoma, with the squamocolumnar junction not visible.

The classifications of normal, precancerous, and cancerous cervical lesions are based on guidelines from the American Society for Colposcopy and Cervical Pathology (ASCCP), which are widely recognized as the standard practice in gynecologic oncology for identifying and managing these conditions [45,46].

### 2.3. Preprocessing and Normalization of Images

For image preprocessing and normalization, the pandas [47] library was used to manage metadata, tensorflow [48] for image processing and model building, and numpy [49] for numerical operations. All images (800 × 600 pixels each) were resized to a standard size of 224 × 224 pixels using TensorFlow’s image processing utilities, ensuring uniformity and facilitating efficient processing [50]. Each image was then converted into an array format suitable for input into the SWIN Transformer and CNN models. To optimize the performance of these models, the images were normalized by scaling the pixel values to the range [0, 1] through division by 255, an important step for speeding up the convergence of deep learning algorithms [51].

#### 2.3.1. Image Selection and Train–Test Splitting

Images were selected based on case numbers and metadata using the pandas library for data manipulation and the os library [52] for file operations. A total of 360 precancerous images from 77 cases, along with an equal number of randomly selected normal cases, were split into train–test sets using pandas. The split was performed at the case level, ensuring that all images from a single case, typically consisting of four or more images, were kept together. The training set included 54 normal and 54 precancerous cases, while the test set comprised 23 normal and 23 precancerous cases, resulting in a roughly 70/30 train–test split. This approach effectively prevented data leakage by ensuring that no images from the same case were split between the training and test sets. It also ensured the model was trained on a representative subset of the data and evaluated on unseen cases for accurate performance assessment [53]. The remaining normal cases, along with the Cancer group were held out as an additional test set. A cross-validation split was conducted at the case level, ensuring that all images from each case were included either in the training or the validation set, preventing overlaps and supporting a more reliable evaluation. In each fold, 11 normal and 11 precancerous cases (representing 20% of the total 54 normal and 54 precancerous cases) were used for validation.

#### 2.3.2. Training Data Augmentation

For the data augmentation process, various techniques were applied to the training images to enhance the diversity of the dataset and improve model generalizations. The transformations included random resizing and cropping to 224 × 224 pixels; horizontal and vertical flipping; color jittering (adjustments to brightness, contrast, saturation, and hue); and random rotations up to 20 degrees. These augmented images were then normalized using the mean and standard deviation of the ImageNet dataset ([0.485, 0.456, 0.406] for the mean and [0.229, 0.224, 0.225] for standard deviations). For the test images, resizing to 256 pixels, center cropping to 224 × 224 pixels, and normalization were applied to maintain consistency during evaluation.

#### 2.3.3. Architectural Flowchart and Mathematical Descriptions of the Hybrid Model

In the SWIN Transformer and CNN hybrid model, CNNs extract local features, while SWIN Transformers capture the global context (Figure 3). These features are integrated and fed into a binary classification layer, which outputs the probability of the image belonging to the precancerous/cancer (positive) class (Figure 3). The model is trained using binary cross-entropy loss to optimize its performance in distinguishing between two classes (“normal” and “precancerous”).

i.Convolutional Neural Network (CNN) Component

The CNN component is responsible for extracting local features from the input image. This process typically involves convolution, activation, and pooling operations:

Mathematical representation



$$\underline{Convolution Operation:}\;\;\;\;X_{l+1}=f(W_{l}*X_{l}+b_{l})$$


$$X_{l}=input\;feature\;map\;at\;layer\;l\;\;\;\;W_{l}=\mathrm{Weight}\;\mathrm{matrix}\;\mathrm{at}\;\mathrm{layer}\;l$$


$$b_{l}=\mathrm{bias}\;\mathrm{at}\;\mathrm{layer}\;l\;\;*=convolution\;operation\;\;f=activation\;function\;(ReLU)$$





$$\underline{Pooling Operation:}\;\;X_{pool}=pool(X_{l+1})\;\;pool = pooling operation (max-pooling)$$



ii.SWIN Transformer Component:

The SWIN Transformer component captures global contextual information using a shifted window-based self-attention mechanism, which helps in understanding the overall structure and relationships within the image.

Mathematical representation
$$\underline{Self-Attention Mechanism:}\;\;\;Attention\left(Q,K,V\right)=softmax\left(\frac{QK^{T}}{\sqrt{d_{k}}}\right)V$$
$$Q=W^{Q}X\left(Query\right)\;\;\;K=W^{K}X\left(Key\right)\;\;\;V=W^{V}X\left(Value\right)$$
$$W^{Q},W^{K},W^{V}=weight\;matrices\;\;\;d_{k}=dimension\;of\;the\;key\;vectors$$
$$\underline{Shifted-Window Mechanism:}\;\;\;Window\;Partition\left(X\right)\to Shifted\;Windows$$

This operation divides the image into non-overlapping windows and applies self-attention within each window, enabling better global-context capturing.

iii.Integration of CNN and SWIN Transformer:

After extracting local features with the CNN and capturing the global context with the SWIN Transformer, the model integrates these features for a final classification.

Mathematical representation


Feature Integration:

$$Z=Concat(Z_{CNN},Z_{SWIN})$$


$$Z_{CNN}=features\;from\;CNN\;\;\;Z_{SWIN}=features\;from\;SWIN-Trasformer$$


$$\underline{Binary Classification Layer:}\;\;\;\hat{y}=\sigma \left(W_{fc}Z+b_{fc}\right)$$


$$W_{fc}=weights\;for\;the\;fully\;connected\;layer$$


bfc=bias for the fully connected layer


bfc=bias for the fully connected layer


$$\hat{y}=predicted\;probability\;for\;the\;precancerous\;or\;cancer\;class$$



iv.Training Process

The model is trained to minimize the binary cross-entropy loss, which measures the difference between the predicted probability and the actual label.

Mathematical representation



$$\underline{Binary Cross-Entropy Loss:}\;\;\;L\left(\hat{y},y\right)=-[ylog\left(\hat{y}\right)+\left(1-y\right)\log\left(1-\hat{y}\right)]$$


$$y=actual\;label\left(0\;or\;1\right)\;\hat{y}=predicted\;probability\;for\;the\;precancerous\;or\;cancer\;class$$





$$\underline{Optimization:}\;\theta =\theta -\eta \nabla_{\theta }L(\hat{y},y)$$


$$\theta =model\;parameters\;\eta =learning\;rate\;\nabla_{\theta }=gradient\;with\;respect\;to\;parameters$$



AdamW optimization [54]: the update rule for AdamW optimization involves additional steps. AdamW (Adaptive Moment Estimation with Weight Decay), being a more advanced optimization algorithm that improves upon traditional gradient descent, includes the following:
Adaptive Learning Rates: AdamW adjusts the learning rate for each parameter individually, based on the first moment (mean) and the second moment (uncentered variance) of the gradients.Weight Decay: AdamW includes a weight decay term that is decoupled from the gradient-based update, helping to regularize the model by discouraging large weights.



$$m_{t}=\beta_{1}m_{t-1}+\left(1-\beta_{1}\right)\nabla_{\theta }L\left(\hat{y},\;y\right)\;v_{t}=\beta_{1}v_{t-1}+\left(1-\beta_{2}\right){(\nabla }_{\theta }L{\left(\hat{y},\;y)\right)}^{2}$$


$${\hat{m}}_{t}=\frac{m_{t}}{1-\beta_{1}^{t}}\;\;\;\;{\hat{v}}_{t}=\frac{v_{t}}{1-\beta_{2}^{t}}$$


$$m_{t}\;and\;v_{t}\;are\;the\;first\;and\;second\;moment\;estimates,\;respectively$$


$$\beta_{1}\;and\;\beta_{2\;\;}are\;hyperparameters\;that\;control\;the\;decay\;rates\;of\;these\;moving\;averages$$


λ is the weight decay coefficient



AdamW is highly appropriate for training CNN and SWIN Transformer hybrid models. Its adaptive learning rates and decoupled weight decay generally lead to better convergence and improved generalizations compared to standard gradient descent.

### 2.4. SWIN Transformer CNN Hybrid Model Training

For the classification task, a SWIN Transformer and CNN hybrid model was designed. The SWIN Transformer architecture, specifically the swin_base_patch4_window7_224 model [19], was chosen as a compromise between performance, computational demands, and data size, given its proven effectiveness in vision tasks and its ability to handle high-resolution images efficiently. The hybrid model combines the SWIN Transformer, which captures global contextual information, with CNN components for feature extraction, followed by a fully connected layer for classification. The output features from the SWIN Transformer undergo global average pooling, are passed through a ReLU [55] activation function, and are finally processed by a fully connected layer for binary classification. Transfer learning was achieved by incorporating pre-trained weights from large image datasets, enhancing the model’s performance, given the limited size of the colposcopy image dataset. The timm (PyTorch Image Models) [56] library provided the pre-trained SWIN Transformer model, while the hybrid model was implemented using PyTorch [57]. This combination of techniques facilitated efficient training and reliable classification performance [58].

#### 2.4.1. Cross-Validation Methodology for SWIN Transformer and CNN Hybrid Model

A five-fold cross-validation (CV) approach was used to optimize the training of the SWIN Transformer and CNN hybrid model. Cross-validation was implemented to ensure the model’s robustness and generalizability across different subsets of the data. Here, the train dataset was split into five folds, with each fold being used once as a validation set, while the remaining folds were used for training. This process was repeated five times, allowing the model to be validated on each fold.

For each fold, the model was trained using the AdamW optimizer with a decaying learning rate, managed by a step learning rate scheduler. The performance of the model was evaluated at the end of each epoch using accuracy, F1 score, and AUC metrics on both validation and the test datasets. The best-performing model, based on test set accuracy, was saved during each epoch. This method ensures that the model is not only trained on a representative subset of the data but also validated on diverse data segments, enhancing its ability to generalize to unseen data. The cross-validation strategy employed here is critical for developing a reliable diagnostic tool that can effectively identify precancerous lesions with high accuracy and consistency.

For hyperparameter tuning, a grid search was performed over a range of learning rates, weight decay values, and batch sizes. The learning rates 0.1, 5e−2, 1e−2, 5e−3, 1e−3, 5e−4, 1e−4, 5e−5, 1e−5, 5e−6, and 1e−6 were tested, along with weight decay values of 1e−2, 1e−3, 1e−4, 1e−5, and 1e−6 and batch sizes of 16, 32, and 64. For each combination of learning rate, weight decay, and batch size, the model was trained using cross-validation, and its accuracy, F1 scores, AUCs, and other metrics were calculated. The hyperparameter set yielding the highest accuracy was selected as the optimal configuration, with the best combination identified as the final set of hyperparameters for training.

#### 2.4.2. Evaluation of the SWIN Transformer and CNN Models on Standard Datasets

The SWIN Transformer and CNN models were evaluated on publicly available benchmark datasets including the MNIST and the Breast Cancer Wisconsin (Diagnostic) dataset, a standard medical image data. The MNIST dataset served as a baseline for image classification tasks, allowing for a comparison with existing models. The Breast Cancer Wisconsin dataset, commonly used in medical image analyses, provided a relevant context to assess the model’s performance. In both evaluations, the developed method demonstrated better performance than what has been previously reported on these datasets, indicating its effectiveness across various image classification tasks.

#### 2.4.3. Model Performance on Test Data

The trained SWIN Transformer and CNN hybrid model was evaluated on the test dataset. Key performance metrics, including accuracy, F1 score, and Area Under the Curve (AUC), were calculated using TensorFlow and the sklearn library [59]. This evaluation provided insights into the model’s ability to generalize to unseen data.

#### 2.4.4. Evaluation of the Held-Out Cancer and Normal Data

An additional performance evaluation was conducted on the held-out dataset comprising cancer and normal images to assess the model’s robustness. This step was vital for understanding the model’s ability to distinguish between normal and cancerous images, particularly those it had not encountered during training. The evaluation on the held-out data using TensorFlow and sklearn demonstrated the model’s effectiveness in distinguishing a normal cervix from any cancerous or other intermediate lesions (such as precancerous), indicating its potential value in diagnostic applications.

### 2.5. Performance Visualization

In this section, we outline key evaluation metrics, many of which are based on the confusion matrix (Table 1), which are typically structured as follows:

Each metric provides insights into the model’s performance across different aspects, which is crucial for evaluating classification models, particularly in medical imaging contexts.

i.Sensitivity (Recall or True-Positive Rate)

Sensitivity measures the model’s ability to correctly identify actual positive cases (e.g., diseased patients). A high sensitivity indicates that the model has few false negatives, meaning it is effective at capturing true positives: $sensitivity=\frac{TP}{TP+FN}$. A model with high sensitivity ensures that most of the actual positive cases are detected, reducing the risk of missing important diagnoses.

ii.Specificity (True-Negative Rate)

Specificity evaluates the model’s ability to correctly identify actual negative cases. It is critical when minimizing false positives: $specificity=\frac{TN}{TN+FP}$. A high specificity means the model is effective in avoiding false positives, which is vital when a false positive can lead to unnecessary follow-ups or treatments.

iii.Positive Predictive Value (PPV or Precision)

The Positive Predictive Value (PPV), also known as precision, indicates the proportion of positive predictions that are actually correct: $PPV=\frac{TP}{TP+FP}$. High precision means that when the model predicts a positive case, it is likely to be accurate, which is particularly useful when the cost of false positives is high.

iv.Negative Predictive Value (NPV)

The Negative Predictive Value (NPV) measures the proportion of negative predictions that are correct: $NPV=\frac{TN}{TN+FN}$. A high NPV suggests the model is reliable at correctly identifying true negatives, reducing the risk of false negatives.

v.Accuracy

Accuracy represents the overall correctness of the model across all classes, considering both true positives and true negatives: $accuracy=\frac{TP+TN}{TP+TN+FP+FN}$. Accuracy provides a general measure of how well the model is performing but may not be sufficient alone, especially in imbalanced datasets, where the model may predict the majority class correctly while failing to identify minority classes.

vi.F1 Score

The F1 score balances precision and recall, offering a single metric that takes both false positives and false negatives into account: $F1\;score=\frac{2\times (precision\times recall)}{precision+recall}$. The F1 score is particularly useful in datasets with class imbalances, as it balances the trade-offs between precision and recall.

vii.Area Under the Curve (AUC)

The AUC-ROC score quantifies the model’s ability to distinguish between positive and negative cases. A high AUC indicates that the model performs well in distinguishing between the classes across all thresholds.

Additionally, the model’s performance was thoroughly examined using a variety of visualizations, including ROC curves, confusion matrices, and bar plots showing AUC and other key metrics. These visual tools offered a clear and detailed representation of the model’s effectiveness, helping to convey the results comprehensively. The visualizations were generated using the matplotlib and seaborn libraries of Python [60,61] and ggplot2, along with other R packages (https://www.r-project.org).

Generally, the confusion matrix and related metrics provide a comprehensive understanding of the model’s strengths and weaknesses across different dimensions. Sensitivity and specificity focus on the model’s ability to correctly classify positive and negative cases, respectively, while PPV and NPV provide insights into the precision of predictions. Accuracy and the F1 score offer a broader measure of overall performance, and the AUC helps assess the model’s discriminative power. These metrics are especially important in medical contexts, where both false positives and false negatives carry significant consequences.

### 2.6. Detailed Steps of Analysis and Code Availability

The Python code for model implementation and the R scripts for generating graphics are located at the GitHub repository https://github.com/Foziyaam/SWIN-Transformer-and-CNN-for-Cervical-Cancer. This repository contains custom scripts and detailed steps of analyses, ensuring reproducibility and transparency.

## 3. Results

### 3.1. Clinical Data Analysis

A total of 197 cases were analyzed and categorized into normal (94 cases), precancerous (77 cases), and cancerous (26 cases) (Figure 4). All samples were deemed adequate for evaluation, with the visibility of the transformation zone (TZ) varying significantly across cases: Type 1 (completely visible) in 114 cases, Type 2 (partially visible) in 33 cases, and Type 3 (not visible) in 50 cases (Figure 5). The distribution of TZ visibility within diagnostic categories showed that normal cases predominantly had Type 1 TZ (79 cases), and precancerous cases distributed across all TZ types, while the cancer cases, except for one, had Type 3 TZ (Figure 5).

The HPV status was available for 196 of 197 cases, with a higher prevalence of HPV positivity in precancerous cases (54 out of 77) and all cancer cases (26 out of 26), compared to normal cases (10 out of 94) (Figure 5). A colposcopy assessment revealed that normal findings, such as original squamous epithelium and columnar epithelium, were more common in normal cases, while abnormal findings, like dense acetowhite epithelium, punctation, and suspicious lesions, were more prevalent in precancerous and cancer cases (Figure 2). Lesions were mostly located inside the T zone, especially in the cancer cases, whereas more advanced or extensive lesions were found outside the T zone.

The Swede score analysis was an important indication of lesion severities, with the normal cases scoring between 0 and 2, precancerous cases between 3 and 7, and cancer cases > 7 (Figure 6a). Management strategies, such as LLETZ for high-grade lesions and punch biopsies for visible lesions, were determined based on lesion severity. Histopathological outcomes confirmed no significant pathological findings in the normal cases, while the precancerous cases showed varying degrees of dysplasia (LSILs, HSILs, CIN1, CIN2, and CIN3), and cancer cases were confirmed with invasive squamous cell carcinoma, adenocarcinoma, and adenocarcinoma in situ (Figure 2).

#### 3.1.1. Correlation Among HPV Positivity, Cancer Status and Transformation Zone

All the 26 cancer cases are HPV positive and predominantly have Type 3 transformation zones (Figure 5). A majority of the precancerous cases are HPV positive and exhibit Type 1, 2, and 3 transformation zones, while normal cases are primarily HPV negative and have mainly Type 1 transformation zones (Figure 5).

#### 3.1.2. Swede Score Distributions

The Swede score distributions varied by cancer status and HPV status. Higher Swede scores were associated with cancer diagnoses and HPV positivity (Figure 6b).

#### 3.1.3. Summary of the Clinical Findings

Key findings of the clinical data analysis include a high prevalence of HPV in both precancerous and cancer cases and significant differences in colposcopy findings across diagnostic categories. Higher Swede scores were significantly correlated with lesion severity and showed good agreement with histopathological outcomes. The visibility of TZs was related to lesion progression, with higher-grade lesions often being associated with less-visible TZs.

These findings highlight the effectiveness of colposcopy evaluations and the relevance of Swede scores in assessing lesion severity. The strong correlation between HPV status and histopathological outcomes further reinforces HPV’s role as a significant risk factor for cervical neoplasia.

### 3.2. Classifications of Colposcopy Images

#### 3.2.1. SWIN Transformer and CNN Hybrid Model Training with Fivefold Cross-Validation

The hybrid model, combining the SWIN Transformer and convolutional neural networks (CNNs), was trained on a dataset consisting of 486 images (from 54 normal and 54 precancerous cases). The model was trained using transfer learning techniques with pre-trained weights for 30 epochs (Figure 7a) with a batch size of 32, a learning rate of 5e−05, a weight decay of 0.05, and a gamma of 0.8, and with a five-fold cross-validation. The model achieved a validation Area Under the Curve (AUC) of 94% (Figure 7b) and an accuracy of 87% (Figure 7c), indicating that it was able to classify most of the validating images into normal and precancerous categories, as shown by the corresponding confusion matrix (Figure 7c). Other metrics that were calculated based on the confusion matrix include validation sensitivity, 0.86; specificity, 0.90; the positive predictive value (precision), 0.92; and the negative predictive value, 0.81, which were derived from the confusion matrix. The training loss decreased sharply with increased epochs (Figure 7a), indicating efficient learning conditions (including optimal hyperparameter combinations).

#### 3.2.2. Trained Model’s Performance on the Test Set 1 (Precancerous vs. Normal)

The performance of the hybrid model was evaluated on a test set of 222 images (from 23 normal and 23 precancerous cases). The model achieved an AUC of approximately 80% (Figure 8a) with an accuracy of 75% (Figure 8b) on the test set. These results demonstrate that the model can correctly distinguish between normal and precancerous colposcopy images in four out of five cases. Additional values of the model performance metrics include sensitivity, 0.75; specificity, 0.75; the positive predictive value, 0.76; and the negative predictive value, 0.74. The performance was tested using the same hyperparameters that were used for training: batch size = 32, epochs = 30, learning rate = 5e−05, weight decay = 5e−02, and gamma = 0.8.

#### 3.2.3. Performance Evaluation on Test Set 2 (Cancer vs. Normal)

To further assess the robustness and generalizability of the model, it was evaluated on the second test set consisting of 233 equivalent numbers of cancer and normal images. The trained SWIN Transformer and CNN classifier model achieved an AUC of 82% (Figure 9a) and an accuracy of 75% (Figure 9b) on the second hold-out test set. This high performance indicates the model’s efficiency in distinguishing normal images from any abnormal images it was not trained on, particularly for cancer images which were not even the same type as that of the original training data. Additional values of the important metrics include sensitivity, 0.72; specificity, 0.80; positive predictive value, 0.85; negative predictive value, 0.65; and F1, 0.78. The same hyperparameters were used for this evaluation as well.

#### 3.2.4. Performance of the SWIN Transformer and CNN Models on Standard Datasets

An evaluation of the SWIN Transformer and CNN models on the MNIST dataset achieved almost perfect classification, correctly classifying the digits from 0 to 9 of the test data, indicating the generalizability of the model. Additionally, an evaluation of the model on the Breast Cancer Wisconsin (Diagnostic) dataset, a modest standard medical image dataset, and the accuracy was above 96%. The high performance of the model on the Breast Cancer Wisconsin dataset provided very high confidence in efficiency and relevance of the model with regard to medical image classifications, providing a relevant context to assess the model’s performance. In both evaluations, the developed method demonstrated better performance than what has been previously reported on these datasets, indicating its effectiveness across various image classification tasks.

#### 3.2.5. Model Performance Summary

The results of this study demonstrate the potential of this hybrid model in cervical cancer diagnosis, with strong performance on validation and independent test sets. The model showed reliable generalizations to new, unseen data, particularly for cancer images not included in the training phase. By combining CNN’s local feature extraction with the SWIN Transformer’s global context processing, the model successfully identified differences between normal, precancerous, and cancerous cases. While the results are promising, variability in image quality and clinical data suggests further refinements may improve the model’s application in early-stage cervical cancer screening. Overall, the SWIN Transformer and CNN hybrid model offers a practical approach for medical image classifications and early diagnoses.

The SWIN Transformer and CNN hybrid model for a precancerous colposcopy classification integrates both architectures to provide accurate and interpretable results. The CNN component focuses on extracting local features, such as edges, textures, and shapes, while the SWIN Transformer captures both local and global contextual information, modeling relationships across the image. These features are combined and processed through a classification layer, effectively distinguishing subtle differences in tissues, which is important for identifying precancerous lesions. The model benefits from transfer learning and data augmentation, allowing it to perform well despite limited data. Techniques like contrastive learning and multi-task learning also help the model concentrate on disease-specific patterns while reducing the influence of confounding factors.

#### 3.2.6. Complementing Traditional Methods

The hybrid deep learning model improves the diagnostic process by providing a consistent, accurate analysis during colposcopy, complementing primary screening tools like cytology (Pap smears) and HPV testing, which can vary in interpretation and have lower sensitivity [15,16]. A human interpretation of colposcopy images is subjective, varying due to the clinician’s experience, fatigue, or bias, which can lead to inconsistent diagnoses. Subtle lesions may be overlooked, and the process is time-consuming and resource-intensive. The hybrid model can potentially address these challenges by offering a standardized, objective analysis, detecting subtle features missed by humans and providing rapid, real-time diagnostic support with better scalability and accessibility, especially in resource-limited settings. It reduces possible human errors, offering advantages in consistency, accuracy, and scalability, and ultimately enhancing the early detection of precancerous lesions.

Particularly, the SWIN Transformer’s ability to model long-range dependencies and capture the global context makes it particularly well-suited for tasks like colposcopy image classification, where subtle differences in tissue structures can be critical. These results suggest that integrating AI-driven diagnostic tools into clinical practice could significantly improve the accuracy and efficiency of cervical cancer screening programs, especially in regions with limited healthcare resources.

## 4. Discussion

Cervical cancer remains a significant public health challenge, especially in low-resource settings where early diagnosis and timely treatment are often hindered by limited access to healthcare. The WHO has highlighted the importance of early detection in reducing cervical cancer-related morbidity and mortality, emphasizing the need for more effective diagnostic tools in these regions [62]. The results of this study, which employed a SWIN Transformer and CNN hybrid model for the classification of colposcopy images, indicate that such advanced machine learning techniques have the potential to mitigate these challenges.

The SWIN Transformer and CNN hybrid model achieved an Area Under the Curve (AUC) of approximately 80% to 82% and an accuracy of 75% on two independent test datasets, a notable achievement given the moderate size of the training data. These performance metrics indicate the model’s effectiveness in distinguishing between normal and precancerous colposcopy images, as well as between normal and cancerous images that were not included in the initial training set. To contextualize this performance, it is worth noting that the US Food and Drug Administration-approved MammaPrint—a 70-gene expression profile test used to assess the risk of breast cancer recurrence in early-stage patients—reports an AUC ranging between 68% and 75% [63,64,65]. This comparison highlights the significance of our model’s results, particularly in the early detection of precancerous lesions, which is important for preventing the progression to invasive cervical cancer, a condition with high mortality rates [4]. The model’s robust performance on unseen data further suggests its potential utility in clinical settings, where generalizability across diverse patient populations is essential.

The model’s ability to generalize well, even when faced with unseen cancer images, indicates its robustness and suggests that it could be a valuable tool in clinical practice to differentiate normal from any other type of lesion abnormalities. Previous studies have shown that deep learning models can outperform traditional diagnostic methods in medical imaging, such as CNNs achieving a dermatologist-level classification in skin cancer diagnosis [9] and superior performance in detecting pneumonia from chest X-rays [66]. The success of the SWIN Transformer CNN hybrid model in this study aligns with these findings, indicating that such models could become a critical component of cervical cancer screening programs, particularly in resource-limited settings, where access to expert cytologists is scarce [67].

### 4.1. Clinical Implications

The improved accuracy and AUC of the hybrid model indicate its potential for clinical applications. Early detection facilitated by such models can lead to timely interventions, reducing the progression to invasive cervical cancer and lowering mortality rates [6,8]. The hybrid model supplements human interpretation with a standardized, algorithm-driven analysis, ensuring the accurate identification of LSILs and HSILs based on specific visual and textural features. This model enhances detection and classification capabilities by providing a nuanced and comprehensive analysis of colposcopy images, integrating detailed image assessments with a broader contextual understanding. The model’s real-time analysis and continuous learning from diverse datasets strengthen its generalizability across different patient populations, making it a reliable tool in various clinical settings. The developed models demonstrate the potential to effectively differentiate normal cervical images from abnormal lesions. This capability extends beyond distinguishing precancerous lesions, as the models can also reasonably distinguish normal images from those of cancer images that were not the same type as that of the train set.

This study highlights the potential of combining SWIN Transformers and CNN models for the early diagnosis of cervical cancer using colposcopy images. If integrated into existing screening programs, this AI-driven approach could standardize the evaluation of colposcopy images, reducing the variability that is often faced in traditional methods like Pap smears [15] and HPV testing [16]. By providing more consistent and accurate diagnoses, such models could help reduce the number of false positives, thereby lowering the psychological and economic burden on patients and healthcare systems alike. This is particularly relevant in low- and middle-income countries, where healthcare resources are often stretched thin, and the burden of cervical cancer is disproportionately high [4].

Moreover, the model’s performance suggests that similar approaches could be applied to other areas of medical imaging. The versatility of the SWIN Transformer, which captures global contextual information, combined with the CNN’s strength in feature extraction, makes this hybrid approach well-suited for a wide range of diagnostic tasks. Future studies should explore the application of this model to other cancers and diseases, potentially extending its use to mammography, lung cancer screening, and beyond. The success of such models could lead to the development of comprehensive AI-driven diagnostic platforms that integrate multiple imaging modalities, further enhancing the early detection and treatment of various conditions [68].

### 4.2. Limitations and Future Directions

Despite these promising results, several challenges remain. One of the most significant barriers to the widespread adoption of AI in healthcare is the need for large, annotated datasets to train and validate these models. While transfer learning and data augmentation can mitigate this issue by leveraging pre-trained models and generating new data, it is important to expose the models to a diverse range of cases to ensure generalizability [21]. For example, validating these findings with larger and more diverse datasets, and integrating additional data sources, such as multi-omics, genomics, and patient history, could significantly enhance the model’s robustness and make it a more comprehensive diagnostic tool [21,29].

In addition, integrating AI tools into clinical workflows will require rigorous validation and adherence to regulatory standards to ensure their safety and effectiveness [69]. The ethical considerations surrounding the use of AI in healthcare, including issues of data privacy, algorithmic transparency, and potential biases, must also be addressed to build trust among clinicians and patients [70].

Looking forward, the integration of much larger multimodal datasets could significantly enhance the model’s diagnostic accuracy and robustness. This holistic approach would provide a more comprehensive understanding of disease progression and could lead to more personalized treatment strategies [71]. The development of explainable AI models, which offer insights into the decision-making process of the algorithms, could further facilitate the adoption of these tools in clinical practice by allowing clinicians to better understand and trust the AI’s recommendations [71].

## 5. Conclusions

This study highlights the potential of SWIN Transformer and CNN hybrid models for the early diagnosis of cervical cancer through colposcopy image classifications. The models demonstrated strong performance, with AUC values ranging between 80% and 82% and with an accuracy of 75% to 78% on independent test datasets. These results indicate the models’ effectiveness in distinguishing between normal, precancerous, and cancerous colposcopy images, particularly in enhancing diagnostic accuracy after abnormal findings from traditional screening methods like cytology and HPV testing.

These models not only improve the early detection of cervical cancer but also offer the potential to reduce unnecessary follow-up procedures and treatments due to false positives, thereby reducing the psychological and economic burden on patients and healthcare systems. Their ability to generalize across diverse patient populations and perform real-time analyses makes them a valuable tool in clinical practice, especially in resource-limited settings with limited access to expert colposcopists.

To maximize the benefits of these models, future research should focus on validating these findings with larger and more diverse datasets. Incorporating additional data sources, such as multi-omics and genomics, could further enhance the models’ robustness and diagnostic accuracy, paving the way for more comprehensive and personalized treatment strategies. Addressing challenges related to dataset diversity, regulatory approval, and ethical considerations, such as data privacy and algorithmic transparency, is also crucial for the successful integration of AI-driven diagnostic tools into clinical workflows.

Developing reliable and accessible diagnostic systems using SWIN Transformer and CNN hybrid models has the potential to significantly reduce the burden of cervical cancer, especially in underserved regions. The findings from this study may guide policy decisions on cervical cancer screening and prevention, ultimately leading to better public health outcomes and an enhanced quality of life for women worldwide.

## Figures and Tables

**Figure 1 diagnostics-14-02286-f001:**
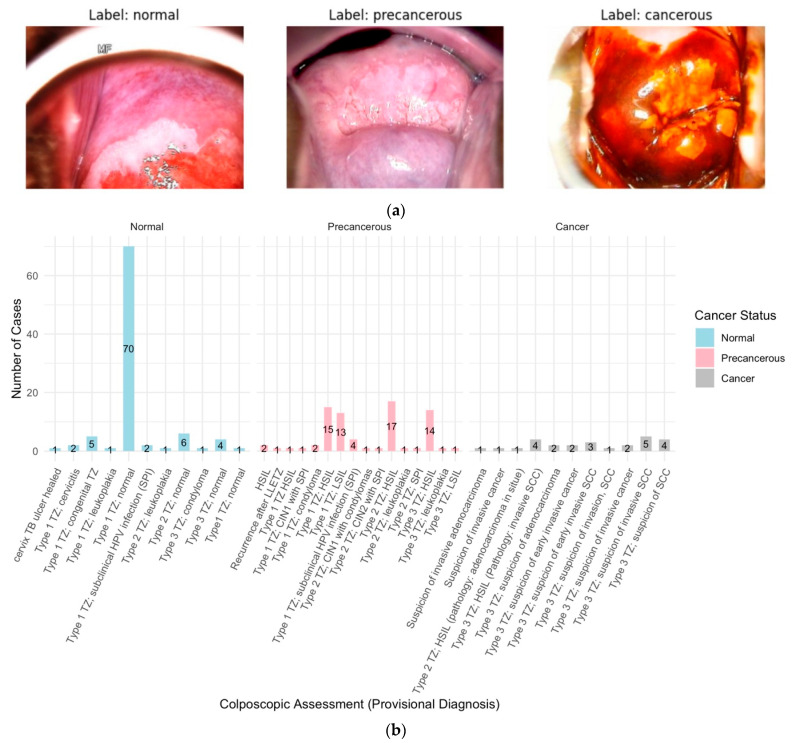
(**a**) Colposcopy images from each of the normal, precancerous, and cancer groups. (**b**) Provisional diagnosis vs. cancer status: The provisional diagnosis during colposcopy, which relies on the clinician’s initial visual and clinical assessment, is important for categorizing cervical lesions as normal, precancerous, or cancerous. An analysis of the provided dataset highlights specific provisional diagnoses associated with each final cancer status. For cases ultimately confirmed as normal, the most frequent provisional diagnosis was “Type 1 Transition Zone (TZ); normal.” In precancerous cases, “Types 1, 2, and 3 TZ; HSIL” and “Type 1 TZ; LSIL” were commonly noted. For cancer cases, “Type 3 TZ; suspicion of invasive squamous cell carcinoma” was the predominant provisional diagnosis.

**Figure 2 diagnostics-14-02286-f002:**
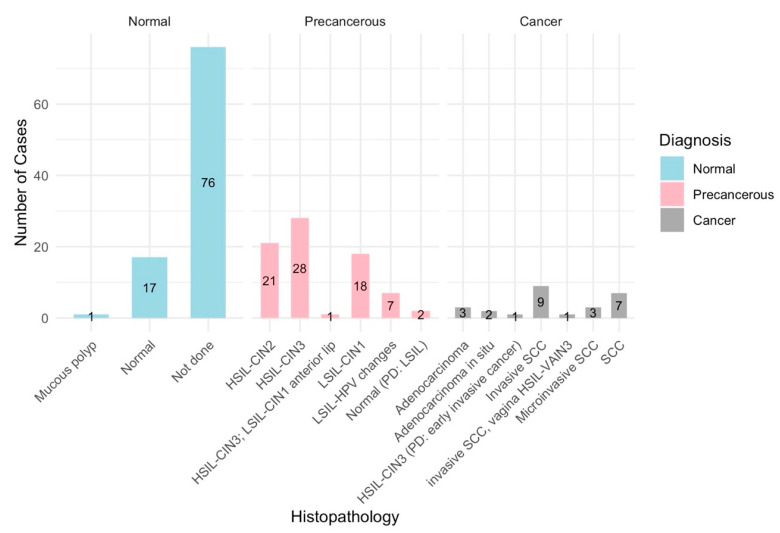
Histopathology vs. cancer diagnosis: A histopathological analysis was used to determine the final diagnosis of cervical lesions identified during colposcopy. In normal cases, histopathology was often not performed, suggesting that the colposcopy assessment alone was sufficient. When histopathology was performed, findings such as CIN1 or the absence of dysplasia supported the normal diagnosis. Precancerous cases were characterized by moderate to severe dysplasia (CIN2, CIN3), low-grade squamous intraepithelial lesions (LSILs), and high-grade squamous intraepithelial lesions (HSILs), indicating varying degrees of abnormality with potential progression to cancer. Cancer cases were confirmed by histopathological evidence of invasive adenocarcinoma, squamous cell carcinoma, or adenocarcinoma in situ. These findings highlight the role of histopathology in accurately diagnosing and categorizing cervical lesions, guiding appropriate patient management and treatment strategies.

**Figure 3 diagnostics-14-02286-f003:**
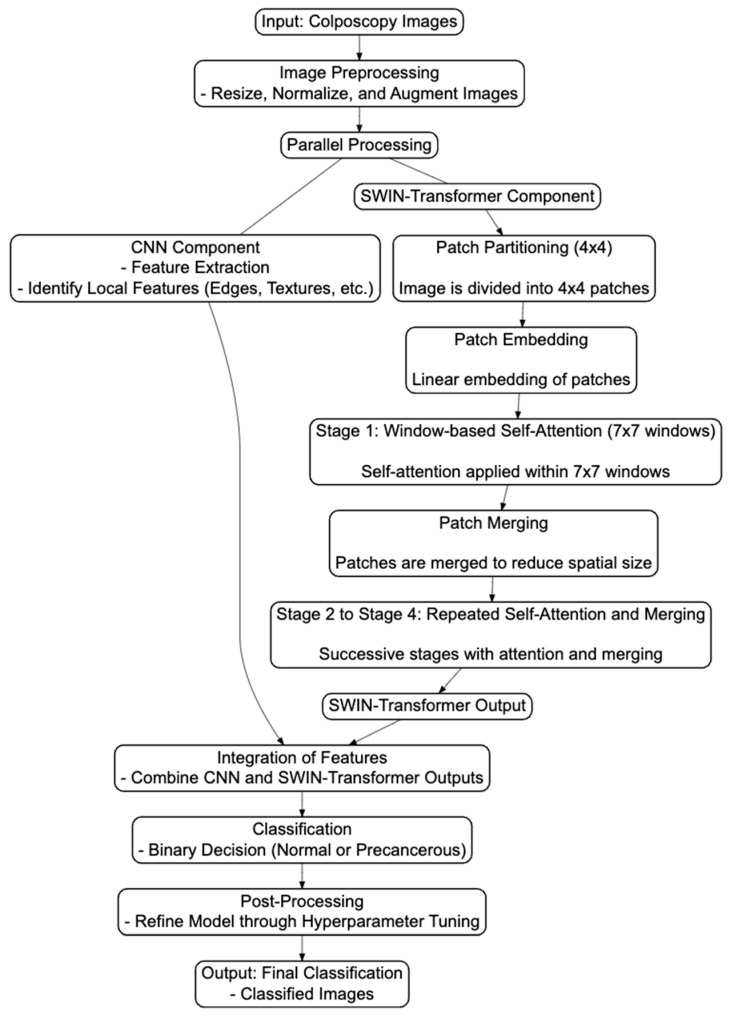
An architectural flowchart illustrating the integrated process where the SWIN Transformer architecture and CNN are combined into a hybrid model for the binary classification of colposcopy images. This diagram highlights the seamless interaction between the two components, demonstrating how they work together to enhance the accuracy of image classification. The CNN and SWIN Transformer processes are parallel, and both outputs are integrated before classification. This flowchart includes specific steps within the SWIN Transformer architecture (swin_base_patch4_window7_224), such as patch partitioning, embedding, window-based self-attention, and merging, before integrating with the CNN outputs. After integration, the process flows through classification, post-processing, and final output generation. Detailed steps and the Python code are available at https://github.com/Foziyaam/SWIN-Transformer-and-CNN-for-Cervical-Cancer.

**Figure 4 diagnostics-14-02286-f004:**
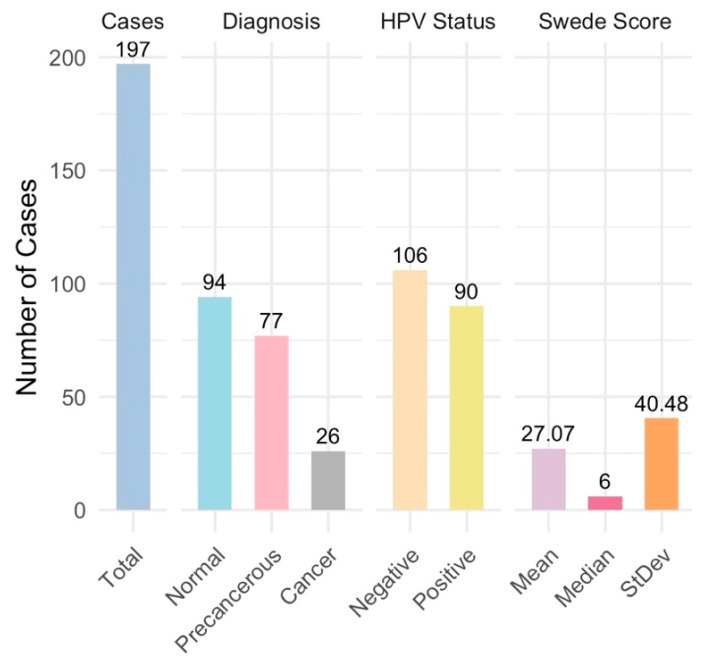
Summary statistics—overall case diagnosis distribution: This summary statistics provides an overview of the distribution of colposcopy cases by diagnosis, Swede score distribution, and HPV status distribution, encapsulating the key aspects of the clinical findings. The distribution of overall case diagnoses indicates a predominance of non-cancer cases, with a significant portion of precancerous and some cancer cases.

**Figure 5 diagnostics-14-02286-f005:**
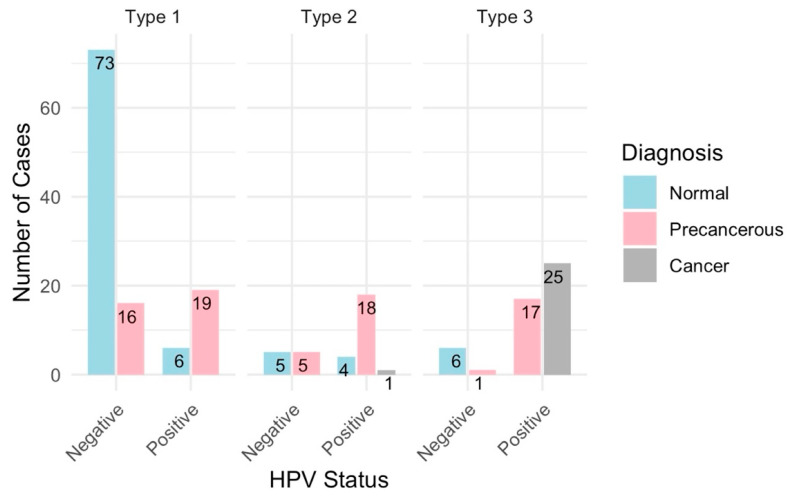
Distribution of cancer diagnoses by HPV status and transformation zone. These are 196 cases, since one of the cases does not have HPV test results.

**Figure 6 diagnostics-14-02286-f006:**
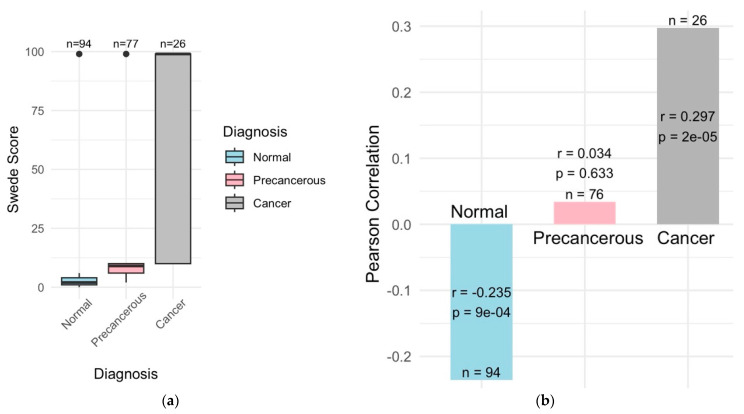
Correlation between Swede scores and cancer diagnosis. (**a**) Normal cases have the lowest-to-no Swede scores and precancerous moderate, while cancer cases have very high Swede scores. (**b**) Swede scores were significantly correlated with cancer diagnosis (r = 0.3 and *p* = 2e−05) and negatively correlated with normal diagnosis (r = 0.2 and *p* = 9e−04) while there was no significant correlation with the precancerous diagnosis.

**Figure 7 diagnostics-14-02286-f007:**
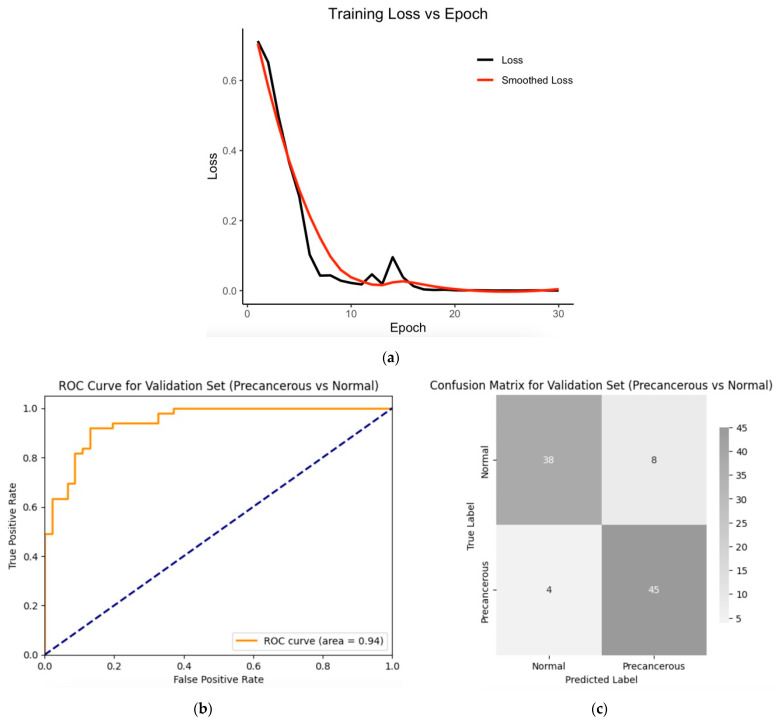
(**a**) Training loss across epochs; fivefold cross-validation metrics (red curve is the smoothing of the actual curve—the black line): (**b**) validation ROC curve for the validation set; (**c**) confusion matrix for the validation set. Validation sensitivity, 0.86; specificity, 0.90; positive predictive value (precision), 0.92; negative predictive value, 0.81; accuracy, 0.87; F1 score, 0.88; and AUC, 0.94.

**Figure 8 diagnostics-14-02286-f008:**
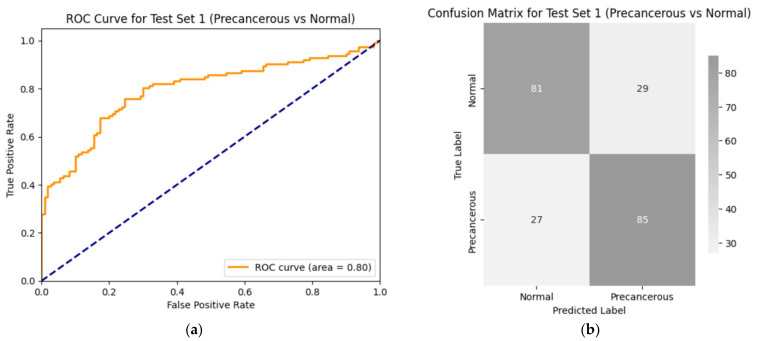
Performance of the trained model on the first test data (precancerous vs. normal): (**a**) ROC curve for test set 1 (precancerous versus normal); (**b**) confusion matrix for the performance of the model on test set 1 (precancerous vs. normal group). The values of the model’s performance metrics include sensitivity, 0.75; specificity, 0.75; positive predictive value, 0.76; negative predictive value, 0.74; accuracy, 0.75; F1 score, 0.75; and AUC, 0.80. The performance was tested using the same hyperparameters that were used for training: batch size = 32, epochs = 30, learning rate = 5e−05, weight decay = 5e−02, and gamma = 0.8.

**Figure 9 diagnostics-14-02286-f009:**
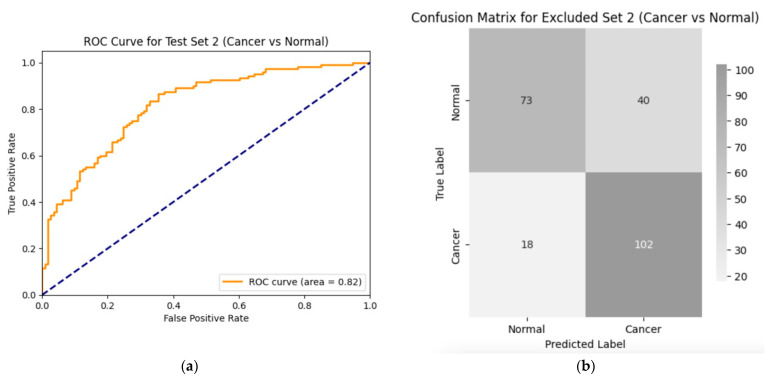
Performance of the second test set (images from cancer and normal cases): (**a**) ROC curve for test set 2 (cancer versus normal); (**b**) confusion matrix for the performance of the model on test set 2 (cancer vs. normal). Values of the important metrics include sensitivity, 0.72; specificity, 0.80; positive predictive value, 0.85; negative predictive value, 0.65; accuracy, 0.75; F1, 0.78; and AUC, 0.82. The same hyperparameters were used for this evaluation as well.

**Table 1 diagnostics-14-02286-t001:** Confusion matrix.

	Predicted Positive	Predicted Negative
Actual Positive	True Positive (TP)	False Negative (FN)
Actual Negative	False Positive (FP)	True Negative (TN)

## Data Availability

Custom Python and R scripts and detailed steps of the analyses are available at the GitHub repository: https://github.com/Foziyaam/SWIN-Transformer-and-CNN-for-Cervical-Cancer.

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
