# Peer review of "Early Cervical Cancer Diagnosis with SWIN-Transformer and Convolutional Neural Networks"

_diagnostics, 2024, doi:10.3390/diagnostics14202286_

Round 1

Reviewer 1 Report

Comments and Suggestions for Authors

1.Improvement

1.1Introduction

1) Update the cervical cancer burden statistics to reflect the most current data available.

2) The part is relatively not abundant, so the importance of colposcopy image classification should be introduced more to reflect the necessity of research.

3) Please clarify the application scenario and research purpose of automatic classification. Is it used to assist colposcopists in diagnosis to improve accuracy and efficiency? The purpose should be to avoid misdiagnosing CIN as normal or normal as CIN. Or application scenario to compensate for the subjectivity of cytological testing (misdiagnosing normal cells as abnormal)? The aim should be to reduce the rate of misdiagnosis.

1.2 Materials and Methods

1) Normal Group should be iodine positive, which cervical epithelium is brownish black or brown.

2) This part is too long and can be cut.

1.3 Results

1) HPV infection is the cause of almost all cervical cancers has been confirmed. The authors should consider whether it is also necessary to demonstrate that The clinical data analysis reveals a significant correlation between HPV positivity and cancer status.

2) Cytology and HPV testing are the primary screening measures, while colposcopy is a referral step for further diagnosis rather than a side-by-side screening measure. The authors should consider whether 3.2.6 is necessary.

3) Clinical implications can be placed into Discussion.

4) It can supplement the correct relationship between LSIL/HSIL and classification.

1.4 Discussion

1) Further discussion of this method can improve the diagnostic efficiency of colposcopists.

1.5 Conclusion

This method cannot compensate for the inadequacy of traditional screening methods. Because according to the screening process of ASCCP, only patients with abnormal initial screening will be referred for colposcopy, so it cannot reduce missed diagnosis.

1.6 Others

1) Please review the paper and unify the format.

2) This should be Figure 3.

3) As mentioned earlier, there is no need to write the full name again.

Reviewer 2 Report

Comments and Suggestions for Authors

- In the abstract section, brief information about the steps of the proposed method is expected to be provided. Information about the dataset and test conditions can be reduced.

- In order not to disrupt the integrity of meaning, it would be appropriate to move the paragraph between lines 97-105 to after line 87.

- The contributions of this method proposed for early diagnosis of cervical cancer should be given in the introduction section as items.

- The steps of the proposed method should be explained in the introduction section, even if only in one sentence. Only the importance of early diagnosis of cervical cancer is mentioned in the introduction section. In addition, information about the architecture of the method should be added to inform the reader at an early stage.

- In addition, the organization of the paper should be added as the last paragraph in the introduction section.

- There are different font sizes in the paper, especially in the materials and methods section. Are these due to the format or were they made by mistake? It is also irregular in form.

- A related studies section examining the literature on the subject should be added.

- A space should be added before [28, 29].

- Figure 2(b) is mentioned in the paper. Figure 2(b) could not be found in the paper. No explanation is given about Figures 1a and 1b.

- There is no mathematical model or formalization of the method proposed in section 2.

- A flow chart of the method proposed in section 2 should be given.

- The text on the x-axis of the figures is very faint, it should be changed to the same text format as the other texts in the figure.

- Figure 8c could not be found in the paper.

- The interpretation of the results obtained in the experimental result section is incomplete. This should not be left only to the discussion section.

- The conclusion section should be detailed.

Comments on the Quality of English Language

- In the sentence "Furthermore, the application of transfer learning have been shown to improve the performance of these models, making them more robust and generalizable across different sizes of datasets [18].", "have" should be corrected to "has".

- The English of the paper should be reviewed. There are grammatical errors.

Reviewer 3 Report

Comments and Suggestions for Authors

This study explores the use of SWIN-Transformer and CNN hybrid models combined with transfer learning to classify precancerous cervical lesions using colposcopy images. The model demonstrates strong performance, with AUCs between 80% and 82% on independent test sets, suggesting its potential for improving early cervical cancer diagnosis. The study highlights the model's effectiveness even with modest datasets and limited data augmentation, emphasizing the need for further validation in more diverse patient populations.

There are some notable strengths of the manuscript,

1.     The study combines SWIN-Transformer and CNN architectures, offering a novel approach to cervical cancer diagnosis.

2.     The manuscript addresses a critical need for improved early detection methods in cervical cancer, with significant implications for patient outcomes.

3.     This study provides a detailed account of model training, validation, and testing, ensuring the study's reproducibility and transparency.

4.     It identifies the need for validation on larger and more diverse datasets, guiding future research efforts in this field.

Few Questions / Comments on the manuscript are as follows,

1.     Cite the library used in section 2.3.1 for Image selection and Train-Test Splitting, the references for library used in this section numpy and os, it would be good to cite the libraries.

2.     Also, the pre-processing mentioned in section 2.3 for preprocessing and normalization of Images, it would be good to know the dimension of the original images acquired from the colposcopy acquisition. Also, citation for libraries for tensorflow and pandas is missing in this section.

3.     Section 2.3.1 for Image selection and Train-Test Splitting, Images from 77 cases were selected for precancerous and similar number of cases were selected for normal. The images from these cases were randomly divided into training and test set. Training and testing images would be from the same subject in this approach.

4.     The approach used for training and testing purpose would involve data leakage between training and validation subset. As the images from same case would be present in both training and testing subset. This would result in inflating the results.

5.     It would be good, if authors would have implemented data splits for training, validation and testing based on case/subjects instead of images.

6.     Section 2.4.2, for Cross-Validation Methodology for SWIN-Transformer and CNN Hybrid Model, Did the author distribute the data based on images? If so, the results will be inflated and will not show true representation of a held-out dataset.

7.     Section 2.4.4, for Model performance on Test data, why did the author did not evaluate the results using other metrics such as sensitivity, specificity, ppv and npv.

8.     The results for the validation set were auroc of 0.94, compared to the test set 1 and test set 2, which did perform at auroc of 0.8 and 0.82. The model performance doesn’t generalize on the given test dataset from the same or the different sample size.

9.     The manuscript mentions extensive grid search for hyperparameter tuning. Can the author provide more insights into the range of hyperparameters explored and how sensitive the model's performance was to these settings?

Comments on the Quality of English Language

The quality of English used in the manuscript was ok

Reviewer 4 Report

Comments and Suggestions for Authors

The authors have proposed a deep learning model for cervical cancer diagnosis. The manuscript is well written however, the following are my concerns: 

The abstract doesn't talk about the novelty and highlights of the proposed model rather it talks about the steps followed. 

The first citation is a website so it is important to mention the date accessed details as per the website citation style. 

Contribution summary is required. Sectionwise outline is missing. 

The literature study provided is insufficient. The authors are expected to study more papers on the transformers and attention based models. 

System architecture diagram is required to understand the complete flow of the proposed work. 

How did Case grouping performed? Based on some standard methods or literature? If so cite them. 

The quality of the images is poor. 

The model architecture of SWIN transformer and CNN model are missing which is crucial. 

Why cross validation? Justification required. 

The loss graph presented lacks legends. 

Plot accuracy graph too for training vs Validation

Hyperparameter tuning details are not available. Present them in a table. 

There is no experimental analysis carried out. Try different hyperparameters and report their results.

Minor concerns:

The font of the manuscript is not consistent. 

Don't write as discussed above or will be discussed next. It is not the standard way to write a manuscript. 

Comments on the Quality of English Language

Minor editing of English language required.

Round 2

Reviewer 1 Report

Comments and Suggestions for Authors

I have noted that the sample size of your study is relatively small, which may limit the generalizability of your findings. In addition, while the development of AI-based colposcopy models is an area of interest, the field has seen significant advancements in recent years. Consequently, the novelty of the approach presented in your manuscript does not sufficiently distinguish itself from existing work in the literature.
I appreciate your understanding and the effort you have put into your research. I wish you success in your future endeavors and hope that you will consider resubmitting a revised version of your work to us at a later date, or that you find a suitable publication venue for your manuscript.

Reviewer 2 Report

Comments and Suggestions for Authors

Although some of the changes suggested in the previous revision were implemented, the following points were not taken into consideration.

- The contribution of the method has not yet been given in the introduction section. It has been stated in the response file to the referee that the contributions have been added. However, the added sentence should be clearly stated. This information is not clear in the response file either.

- In the previous revision, it was suggested to add a related studies section. The authors did not take this into consideration.

- The resolution of the newly added Figure 3 is very low.

- The experimental results have not yet been sufficiently detailed and analyzed.

Comments on the Quality of English Language

The language quality of the paper is average.

Reviewer 4 Report

Comments and Suggestions for Authors

The authors have addressed my concerns. 

Round 3

Reviewer 2 Report

Comments and Suggestions for Authors

The authors have made the necessary revisions, taking into account the issues I suggested. I thank them.